# Effect of 3D Printing Process Parameters and Heat Treatment Conditions on the Mechanical Properties and Microstructure of PEEK Parts

**DOI:** 10.3390/polym15092209

**Published:** 2023-05-06

**Authors:** Honglei Zhen, Bin Zhao, Long Quan, Junyu Fu

**Affiliations:** College of Mechanical and Vehicle Engineering, Taiyuan University of Technology, Taiyuan 030024, China; zhenhonglei0169@link.tyut.edu.cn (H.Z.); zbtyut@126.com (B.Z.); quanlong@tyut.edu.cn (L.Q.)

**Keywords:** poly-ether-ether-ketone (PEEK), 3D printing, printing parameters, heat treatment, mechanical properties

## Abstract

Fused deposition modeling (FDM) processed Poly-ether-ether-ketone (PEEK) materials are widely used in aerospace, automobile, biomedical, and electronics industries and other industries due to their excellent mechanical properties, thermal properties, chemical resistance, wear resistance, and biocompatibility, etc. However, the manufacture of PEEK materials and parts utilizing the FDM process faces the challenge of fine-tuning a list of process parameters and heat treatment conditions to reach the best-suiting mechanical properties and microstructures. It is non-trivial to make the selection only according to theoretical analysis while counting on a vast number of experiments is the general situation. Therefore, in this paper, the extrusion rate, filling angle, and printing orientation are investigated to adjust the mechanical properties of 3D-printed PEEK parts; then, a variety of heat treatment conditions were applied to tune the crystallinity and strength. The results show that the best mechanical performance is achieved at 1.0 times the extrusion rate, varied angle cross-fillings with ±10° intervals, and vertical printing. Horizontal printing performs better with reduced warpage. Additionally, both crystallinity and mechanical properties are significantly improved after heat treatment, and the best state is achieved after holding at 300 °C for 2 h. The resulting tensile strength is close to 80% of the strength of injection-molded PEEK parts.

## 1. Introduction

The emergence of additive manufacturing (AM) technology has replaced subtractive machining, injection molding, and casting for producing many complex industrial products, since R&D (Research and Development) and production cycles are shortened, material utilization is more efficient, and multi-piece structures can be consolidated to produce integral structures [1]. The above benefits are especially obvious for small batch production. Nowadays, a number of additive manufacturing methods have emerged, such as FDM (fused deposition modeling), SLA (stereolithography), SLS (selective laser sintering), DED (directed energy deposition), DIW (direct ink writing), LOM (laminated object manufacturing), etc. [2,3]. Decision making with respect to material–process selection is recognized as a critical issue and has been addressed using knowledge-based approaches [4]. Amongst the above processes, FDM has been well-known for its specialty in producing plastic parts [5], and compared to other plastics-oriented AM processes, FDM can be conducted with less expensive equipment, simple operations, and high stability [6]. As reported, FDM accounts for about 30% of the installed AM systems worldwide [7].

The materials used in the FDM process have the characteristics of fusibility, adhesion, stability, fluidity, low shrinkage, etc. [8,9]. When using these materials to print parts, it is necessary to consider both the characteristics of the material itself (such as melting point, strength, etc.) and the appropriate FDM equipment (such as the maximum heating temperature of the nozzle, the build chamber temperature, the maximum printable size, etc.) [10,11,12]. A variety of materials is available for FDM manufacturing [13], such as ABS (acrylonitrile butadiene styrene), PLA (polylactic acid), PC (polycarbonate), synthetic rubber, and other materials with low melting points [14]. The mechanical properties of these materials are weak, and the application scenarios are limited [15]. The only exception is PEEK (an aromatic crystalline thermoplastic polymer) [16], which is known to be stiff, strong, lightweight, corrosion resistant, and biocompatible [17,18], and these characteristics make it preferred by aerospace, biomedical, and electronic industries and automotive industries [19,20,21].

To make the best use of Poly-ether-ether-ketone (PEEK) materials, it is important to understand the mechanical properties of the 3D-printed PEEK parts [22,23,24,25,26]. There are a variety of factors that affect the FDM printing effect, including nozzle diameter, extrusion temperature, ambient and platform temperatures, filling angle, layer thickness, printing speed, extrusion multiplicity, printing orientation, etc. [27]. Wu et al. [28] conducted tensile, compression, and bending tests on 3D-printed parts with different layer thicknesses and filling angles. It was found that both the layer thickness and filling angle have significant effects on the mechanical properties, concluding that the best mechanical properties were obtained at a layer thickness of 0.3 mm and a filling angle of 0°/90°. Wang et al. [16] investigated the effects of the nozzle diameter, printing speed, and extrusion speed on the mechanical properties of PEEK. It was found that a 0.4 mm diameter nozzle produces higher-strength samples and simultaneously better surface quality. The microscopic characterization images showed that the lower printing speed leads to fewer printing defects and stronger interlayer and interfilamentous bonds. The authors also performed a temperature distribution simulation and found that the viscosity and flow rate of PEEK were optimal at the filament feeding speed of 4 mm/s.

As can be seen from the above literature survey, tremendous experiments have been conducted to explore the FDM process parameters for PEEK and its composites, and some preferable printing setups have been derived. However, there are still some undisclosed parameter combinations that trouble 3D printing practices. For instance, with respect to extrusion speed investigation, Wang et al. [16] carried out simulation analyses and experimental verification of the extrusion speed, the printing speed used in the experiment is relatively slow (only 20 mm/s), and changing the extrusion speed of the commercial 3D printer is difficult. In this regard, we propose a method to adjust the extrusion speed for higher printing speeds and various types of 3D printers: The extrusion speed can be controlled by changing the extrusion rate, which makes our control of the extrusion speed simpler and more convenient. In addition, the study of the filling angle is not comprehensive. Wu et al. [28] investigated the influence of filling angles as well as layer thickness on the mechanical properties of PEEK parts. It was found that the samples built with the 0.3 mm layer thickness and raster angles of 0°/90° have the greatest strength. Ziemian et al. [29] compared the tensile strength of ABS samples filled at 0°, 45°, 90°, and ±45° with injection-molded counterparts; Ahn et al. [30] compared the tensile properties of 0°, ±45°, 0°/90°, and 90° infills. They only performed single-angle filling or double-angle repeated cross-filling. There still is much room for further fine-tuning angle combinations for better properties, and continuous filling angle optimization remains an open problem [31,32]. Filling each layer at different angles is beneficial to weaken the influence of the anisotropy of 3D-printed samples so that all directions can withstand greater stress. In addition, based on the study of interval angle filling, it is necessary to study the printing direction.

The 3D printing of PEEK filaments requires high temperatures, especially for the 3D printer’s ambient temperature, which has a vital effect on the forming difficulty, mechanical properties, and crystallinity [33,34]. Hua [35] investigated the effect of temperature on the shape memory properties of 3D-printed PEEK and focused on the role of extrusion temperature and printing platform temperature. Via experimental analyses, it was found that the nozzle extrusion temperature has little impact on the shape memory performance, while the platform’s temperature has a strong influence. The recovery rate and fixation rate are enhanced with the printing platform’s temperature increment from 90 to 130 °C, and the degree of crystallinity increased from about 19% to about 23%. Yang et al. [34] raised the 3D printer’s ambient temperature from 25 °C to 200 °C. The crystallinity of the prepared sample increased from 17% to 31% with the increase in temperature, and tensile strength increased from less than 60 MPa to 84 MPa. As can be seen from the above literature survey, the heat treatment of PEEK parts will also affect crystallinity and mechanical properties. The author also compared the effects of different heat treatment methods (tempering, annealing, quenching, etc.) on crystallinity. Although different methods have diverse effects on crystallinity, the crystallinities of PEEK parts after heat treatment improved. Yang et al. [36] prepared CF/PEEK (short carbon-fiber-reinforced Poly-ether-ether-ketone)-fabricated parts at 20 °C ambient temperature. The crystallinity of the parts reached 34–36% after holding at 300 °C for 2 h, and the bending strength increased by 147.8%, which achieved 243.2 MPa. It can be seen that the ambient temperature and heat treatment temperature are closely related to the performance indexes of PEEK parts. The effect of various heat treatment parameters on crystallinity is different. In order to separately study the effect of the holding phase on crystallinity, excluding the interference of the heating and cooling rates, we preheated the oven to reach the preset temperature. Then, the PEEK parts are placed in the oven for heat preservation, and heat treatment is carried out by air cooling after reaching the preset time. Then, the mechanical performance of PEEK parts after heat treatment was measured and analyzed.

In this paper, the tensile and bending properties of FDM-printed PEEK parts are investigated by further classifying the extrusion speed, filling angle, and printing orientation. In addition, in order to improve the mechanical properties of PEEK parts printed using low chamber temperatures (without providing an auxiliary heat source for the chamber environment), the effect of the holding stage on the crystallinity was studied, and PEEK parts were heat treated by the above heat treatment method.

## 2. Experimental Preparation

### 2.1. Printing Parameter Settings

The material extrusion speed, filling angle, and printing orientation are the three investigated factors, and the designed experiments are shown in Table 1. For convenience in understanding, the extrusion speed is expressed as the extrusion rate. The feasible range for the extrusion rate is determined via tests. When the extrusion rate reaches 1.25 times, the heating block temperature drops severely, and the PEEK filament cannot be fully melted, which tends to block the nozzle. Therefore, the experiments were conducted to explore extrusion rates set at 1.2 times and below.

Details of the fill angle and print orientation are explained as follows.

Filling angle

Intervals ±10°: 0°, ±10°, ±20°, ±30°, …, 90°, ±100°, …, 180°.

Intervals ±20°: 0°, ±20°, ±40°, ±60°, …, ±160°, ±170°, 180°.

Intervals ±30°: 0°, ±30°, ±60°, 90°, ±120°, ±150°, 180°.

The filling angles mentioned previously are further illustrated in Figure 1 (±10° as an example). The filling method with an interval of ±10° filling is an example. The first layer is filled at 0°, the second layer is filled at 10°, the third layer is filled at −10°, the fourth layer is filled at 20°, and so on. Similarly, when the interval is ±20°, the first layer is filled at 0°, the second layer is filled at 20°, the third layer is filled at −20°, the fourth layer is filled at 40°, and so on.

2.Extrusion rate

The extrusion rate is defined as the ratio of the actually extruded material’s volume relative to the default value of the 3D printer.

3.Printing orientation

Horizontal printing refers to the printing orientation towards which the sample faces when laying down on the platform with its largest flat surface as the contact, as shown in Figure 2a. Vertical printing refers to the printing orientation with the longitudinal side surface contacting the platform, as shown in Figure 2b.

FDM-printed materials are anisotropic with respect to stiffness and strength so that the influence of different printing orientations on mechanical properties can be investigated.

The 3D printer used is BMF-450, and it is manufactured by China Jinan Dadun Technology Co., as shown in Figure 3.

### 2.2. Heat Treatment Settings

The glass transition temperature of PEEK is 143 °C, and the melting temperature is 343 °C [37]. Therefore, the temperature range of the heat treatment needs to be controlled between these two temperatures. Three temperature levels of 150 °C, 200 °C, and 300 °C are selected for investigating heat treatment effects in combination with the heat treatment times of 0.5 h, 1 h, and 2 h, respectively. The specific setups are given in Table 2. Three experimental samples will be prepared for each group for reproducibility.

The heating equipment for the heat treatment of PEEK parts in this experiment comprises the 101-2B electric hot blast constant temperature drying box (as shown in Figure 4), which is provided by China Foshan Jinshen Mechanical and Electrical Equipment Co., Ltd. The heat treatment conditions adopted in this research were preheating the oven to a high temperature and then placing the parts in the oven and holding for a period of time; afterward, the parts are cooled in open air.

### 2.3. Experimental Test Models and Test Method

Dumbbell tensile specimens (ISO 527-2: 2012) and bending specimens (ISO 178: 2010) are prepared according to the shape and size, as shown in Figure 5. All mechanical property tests will be performed on the electronic universal material testing machine (INSTRON-5900, Instron, Boston, USA). The loading speed of the tensile test and bending test is 2 mm/min.

Microscopic characterization will be performed by field emission scanning electron microscopy (SEM) (Zeiss-EVO-10, Carl Zeiss AG, Oberkochen, Germany), and gold spraying was performed prior to testing to improve the electrical conductivity of samples. The enthalpy of the material will be measured by differential scanning calorimetry (DSC) (Mettler-DSC3, Mettler Toledo, Zurich, Switzerland). Nitrogen protection is required during the test, and the flow rate is 50 mL/min. The enthalpy absorbed by PEEK during heating and melting was measured by increasing the heating rate from room temperature to 400 °C at 10 °C/min. The ratio of the measured enthalpy to the theoretical enthalpy of the 100% crystallization of PEEK materials (130 J/g) [34] will be used as the degree of crystallinity.

## 3. Experimental Results and Discussion

The experimental results are divided into two parts: one is the test results related to the printed parameters, and the other is the test results related to the heat treatment.

### 3.1. The Impact of Printing Parameters on Performance

#### 3.1.1. Effect of Extrusion Rate on Mechanical Properties

In this experiment, five different levels of extrusion rates were tested, and the weight of the as-printed samples was measured. The strength results are summarized in Figure 6a, and the bending strength results are presented in Figure 6b. Both average values and error bars are plotted [38]. We can clearly see that the tensile strength for the normal extrusion rate (1.0 times), which is 69.35 MPa, is higher than the samples with altered extrusion rates; bending strength increases with an increasing extrusion rate from 144.16 MPa to 160.08 MPa, but the marginal improvement is minor when exceeding the normal rate. Comparing the samples with the highest extrusion rate (1.2 times) with the ones from normal extrusion, tensile strength decreases by 16.87% but bending strength increases by 11.04%. Hence, both extrusion rates seem to be good options for the exact application scope. However, the decrease in tensile strength at 1.2 times is higher than the increase in bending strength. Therefore, extrusion at 1.0 times is a better choice.

From the average weight curves of both figures, when the extrusion rate is over 1.0 times, the sample weight increases at a slower speed compared to the scenario in which the extrusion rate is below 1.0 times, resulting in a non-linear relationship between the sample weight and extrusion rate. The reason behind this phenomenon is that when the extrusion rate proceeds beyond 1.0 times, the part’s height is greater than the designed value, which blocks the filaments from the 3D printer to produce the filament smoothly, and it forms a buildup at the extrusion’s head, causing the filament’s output to decrease and not reach the theoretical weight.

Figure 7 shows the bending samples at different extrusion rates. When extruding below the normal rate, some gaps are visible because of insufficient infills. At 0.8 times the extrusion rate, the outer profile is separated from the zigzag infill due to insufficient bonding. When extruding beyond the normal rate, surface quality deteriorates due to the over-accumulation of materials, leading to inappropriate layer height.

Figure 8 shows the surface microscopic images of the samples printed at the extrusion rates of 0.8 times and 0.9 times. The gaps between filaments can be clearly identified, especially for the 0.8 times extrusion, which explains the poor mechanical strength. Figure 9 shows the microscopic images of the sample’s cross-section printed at extrusion rates of 1.0–1.2 times. With an increasing extrusion rate, the layers are better blended with gradually added stacking signs, but printing defects (interlayer gaps, air pores, etc.) appear more frequently, resulting in damaged filaments in the tensile orientation and causing compromised tensile strength. On the other hand, the high extrusion rate leads to better interlayer bonding (less delamination) and thus enhanced three-point bending strength since the bending strength of horizontally printed samples is mainly influenced by the interlayer bonding level.

#### 3.1.2. Influence of Filling Angle on Mechanical Properties

As shown in Figure 10, the tensile and flexural strengths of the widely used repeated ±45° filling are compared with our proposed varied angle cross-filling. It was found that the tensile strength improved with varied angle cross-fillings for both ±10° intervals and ±20° intervals from 64.87 MPa to 69.35 MPa and 67.14 MPa, respectively; the highest bending strength is achieved with both repeated ±45° filling and varied angle cross-fillings with ±10° filling intervals, reaching over 140 MPa. The differences in weight among the above four samples are not significant.

Figure 11 shows the cross-section microscopic SEM images for the cross-infilled PEEK samples filled at intervals of ±10°, ±20°, and ±30°. The cross-sections look similar with visible interlayer interfaces, firmly bonded filaments, and rare printed defects. Wu et al. [28] compared the tensile strength and bending strength of PEEK parts filled at 0°, 30°, and 45°. The sample filled at 0° was parallel to the load direction and was not subjected to shear stress, so its tensile strength was the highest. Weak interlayer bonding is the main reason for the failure of bending samples such that the differences in tensile strength are mainly due to the filling angles. Higher tensile strength can be reached if the filaments and loading orientation are better aligned. In the case of three-point bending, bending strength is affected by both the interlayer bonding effect and the alignment of filaments relative to the loading orientation; hence, the repeated ±45° filling and varied angle cross-fillings with ±10° intervals perform better.

#### 3.1.3. Effect of Print Orientation on Mechanical Properties

Figure 12 shows the tensile strength and bending strength of sample parts at two different printing orientations. It can be observed from the figures that the printing orientation has little effect on the tensile strength, but the material consumption of vertical printing is higher than horizontal printing; printing orientation has a significant influence on the bending strength, jumping from 122.53 MPa (horizontal printing) to 156.84 MPa (vertical printing) at an increased rate of 27.83%. Ding et al. [39] studied the printing orientation, and it was found that the interlayer bonding strength is the main reason affecting the bending strength. To disclose the reason for this phenomenon, the bending failure details of the horizontally and vertically printed samples are presented in Figure 13. It can be observed that the strength is greatly affected by the interlayer bonding effect, and the horizontally printed sample has weaker interlayer bonding since the adhesion between layers is less solid than inlayer filament bonding. The sample part’s load-bearing capacity decreases drastically when interlayer debonding was initiated.

#### 3.1.4. Discussion of the Printing Parameters

As shown in Table 3, according to the test results of mechanical properties with different printing parameters, we analyzed the ratio between the maximum and minimum values of the tensile strength and bending strength in each printing parameter. It can be observed in the table that the difference between the maximum and minimum mechanical properties of the extrusion rate is the largest compared with the mechanical properties of the filling angle and the printing orientation. Compared with the minimum strength, the tensile strength increased by 118.22%, and the bending strength increased by 183.43%; the extrusion rate is the most influential parameter of performance. Compared with the extrusion rate, the filling angle and printing orientation have little influence on mechanical properties, and the difference in tensile strength between different printing orientations is only 0.25% Therefore, among the three parameters in this experiment, the extrusion rate is the most influential one with respect to mechanical properties.

Combined with experimental test results and related theories, we obtained the best combination of printing parameters using the following analysis:(1)When the extrusion rate is lower than the normal extrusion rate, the filling in the filament is not full, and there is a large gap between filaments (shown in Figure 8), which makes PEEK parts fail easily due to poor bonding effects, resulting in a lower extrusion rate and worse strength. This is the main reason why the tensile strength is only 31.78 MPa, and the bending strength is only 56.48 MPa when the extrusion rate is 0.8 times. When the extrusion rate is higher than the normal extrusion rate, with the increasing extrusion rate, the layers are better blended with gradually added stacking signs, but printing defects (interlayer gaps, air pores, etc.) appear more frequently, making the filaments damaged in the tensile orientation and resulting in compromised tensile strength (shown in Figure 9). At 1.0 times extrusion, the tensile strength reached 69.35 MPa, which was 20.29% higher than that at 1.2 times extrusion. Bending strength is significantly affected by the interlayer bonding force. Because of the accumulation of the filament, the boundary between the layers is blurred, resulting in the following: The higher the extrusion rate, the greater the bonding force between layers, and the greater the bending strength, reaching a maximum of 160.88 MPa, which is 11.04% higher than that of the 1.0 times extrusion. Although the bending strength at 1.2 times extrusion is higher than that at 1.0 times extrusion, the increase in bending strength is less than the decrease in tensile strength; thus, 1.0 times is the best extrusion rate.(2)When filled at intervals of ±10°, the sum of the angles between the filling angle of the PEEK filament and the tensile direction is smaller than the sum of the angles of other filling angles. The filaments were subjected to both tensile and shear stress: The greater the angle, the greater the shear stress. Therefore, tensile strength is the greatest when filled at intervals of ±10°, reaching 69.35 MPa. The strength of the bending sample is mainly affected by the interlayer bonding force, but by using test data, combined with Wang et al.’s [28] research on the filling angle, it was found that the filling angle also affects bending strength. Therefore, the bending strength of PEEK parts filled with interval ±10° is also the best, reaching 144.16 MPa.(3)Horizontally and vertically printed PEEK parts have the same filling angle, so their tensile strength is close. In the bending test, the interlayer bonding strength has little effect on the vertical printing sample. Therefore, the bending strength of the vertical printing sample can reach 156.84 MPa, which is 28.00% higher than that observed in horizontal printing.

According to the experimental analysis, we can draw the following conclusion. At 1.0 times the extrusion rate, varied angle cross-fillings with ±10° filling intervals and vertical printing comprise the best combination.

### 3.2. The Impact of Printing Parameters on Warpage Effects

Among the three printing parameters investigated in this paper, printing orientation has the greatest influence on the warpage effect of PEEK parts. Figure 14 shows that there are different degrees of warpage at both ends of the vertically printing PEEK parts during the process, which results in parts that are far from the expected design’s size. Additionally, there is a certain probability of printing failure due to warpage. The horizontally printed PEEK parts have almost no warpage.

According to the mechanical experimental results, vertically printing PEEK parts is a good choice. However, considering the forming difficulty and the warpage effect of printed PEEK parts, horizontal printing is a better choice. Therefore, comprehensively, the best combination of printing parameters is an extrusion rate of 1.0 times, varied angle cross-fillings at ±10° filling intervals, and horizontal printing.

### 3.3. Heat Treatment Effect

In this section, the samples are printed again using the optimum printing parameters obtained from Section 3.2 and then heat-treated under different conditions to disclose their effect on mechanical performances. Varying heat treatment conditions are listed in Table 2.

#### 3.3.1. Heat Treatment Effect on Mechanical Properties

Figure 15 shows the tensile strength, tensile modulus, and bending strength of the sample parts after the heat treatment, respectively; it can be observed that tensile strength, bending strength, and elastic modulus are enhanced with increasing heat treatment temperature and time. Compared with samples without heat treatment, the average tensile strength increased by 11.41%, reaching 77.26 MPa (300 °C, 2 h), which is close to 80% of conventional injection-molded PEEK parts [34]. The tensile modulus increased from 1.65 GPa to 2.18 GPa (300 °C, 2 h). The bending strength increased from 144.16 MPa to 172.98 MPa (300 °C, 2 h), with an increase of 20%. Additionally, as shown in Table 4, we analyzed the effect of temperature on mechanical properties when holding for 0.5 h and the effect of holding times on mechanical properties when the heat treatment temperature is 300 °C. The tensile strength and bending strength increased by 4.80% and 6.20% when the heat treatment temperature increased from 150 °C to 300 °C. When the heat treatment temperature is 300 °C and the holding time increases from 0.5 h to 2 h, the tensile strength and bending strength increase by 4.07% and 3.50%, respectively. It can be observed that heat treatment temperature has a greater influence on the mechanical strength of parts compared to the heat treatment time.

#### 3.3.2. Crystallinity Testing of Samples Featuring Different Heat Treatment Conditions

We also tested the crystallinity of samples featuring different heat treatment conditions, and the results are shown in Table 5. As observed in the data, crystallinity increased from 16.10% to the maximum of 28.70% after heat treatment. Crystallinity increased with higher heat treatment temperatures and extended heat treatment times, and the phenomena are consistent with the trends of tensile and bending property variations. Therefore, heat treatment is suggested as a necessary post-processing procedure for improving mechanical properties and crystallinity such that the benefits of FDM 3D printing can be magnified with enhanced fundamental PEEK material properties.

After isothermal crystallization at different heat treatment conditions, both tensile strength and bending strength have been improved significantly. The reason is that above PEEK glass transition temperature, the molecular chain of PEEK is rearranged and driven by inputted thermal energy so that the short chains produced by the FDM process are united to form long chains, and the crystalline region in the material increases, forming more crystal structures, which enhanced the crystallinity of parts [16,34]. The increase in crystallinity makes the interwoven parts between the layers form longer chains and crystal structures, thus improving the layer–layer bonding strength. Additionally, heat treatment releases internal residual stresses, also contributing to improved tensile and bending strength, as well as the elastic modulus. As shown in Figure 16 and Figure 17, the strength of the sample parts after heat treatment all increases regardless of processing temperatures, but toughness deteriorates at the same time since no necking stage was observed and only a brittle fracture occurred. The reason for this phenomenon is that increased crystalline regions hinder the movement and sliding of molecular chains, making the material more brittle.

## 4. Conclusions and Future Work

### 4.1. Conclusions

(1)The best comprehensive mechanical properties of printed PEEK parts can be obtained at varied angle cross-fillings at ±10° filling intervals, using a 1.0 times extrusion rate and vertical printing. The heat treatment at 300 °C for 2 h is suggested for post-processing, since mechanical properties can be further improved with the tensile strength reaching 77.26 MPa and the bending strength exceeding 170 MPa.(2)The mechanical properties under different extrusion rates were tested, and the mesoscale structures of the printed parts were analyzed. The tensile strength reached 69.35 MPa when the extrusion rate is 1.0 times. Although the bending strength improved with the increase in extrusion rate, the improvement was not as substantial as the decrease in tensile strength. Therefore, a normal extrusion rate of 1.0 times is suggested.(3)Compared with other filling angles, the tensile strength with varied angle cross-fillings at ±10° intervals improved significantly, reaching 69.35 MPa.(4)Vertically printed PEEK parts have better mechanical properties than horizontally printed ones, and bending strength increased by 27.83%, reaching 156.84 MPa. However, vertical printing will result in poor part quality, such as excessive warpage. Therefore, Considering the forming difficulties and the mechanical properties comprehensively, horizontal printing is a better choice.(5)Heat treatment is an effective way to improve crystallinity, the highest crystallinity reached 28.70%, and it is a necessary post-processing procedure since PEEK parts printed at a low chamber temperature have poor crystallinity, which would not warrant sufficient mechanical properties.

### 4.2. Future Work

The heat treatment conditions adopted in this research were preheating the oven to a high temperature and then placing the parts in the oven and holding it there for a period of time; afterward, the parts were cooled in the open air. The influence of heating and cooling rates on crystallinity has not been investigated, and future work will concentrate on further tuning heat treatment conditions [40].

## Figures and Tables

**Figure 1 polymers-15-02209-f001:**
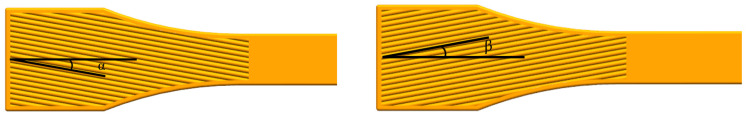
Illustration of two layers with different filling angles: α = 10° and β = −10°.

**Figure 2 polymers-15-02209-f002:**
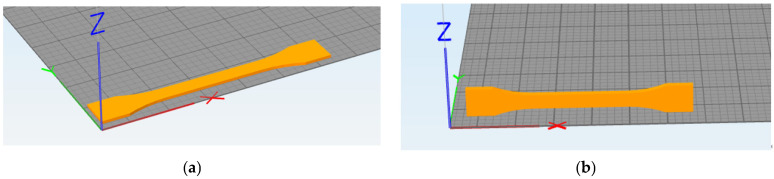
(**a**) Horizontal printing demonstration; (**b**) vertical printing demonstration.

**Figure 3 polymers-15-02209-f003:**
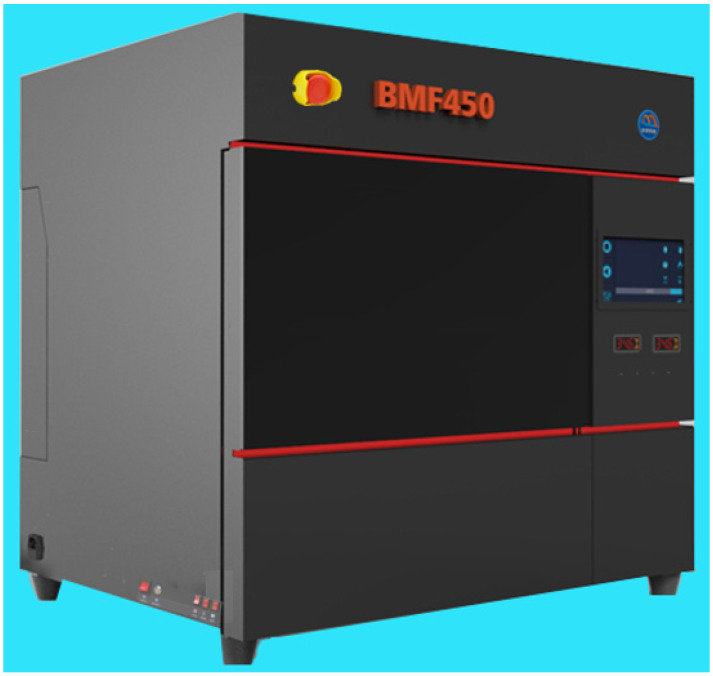
BMF-450 3D printer.

**Figure 4 polymers-15-02209-f004:**
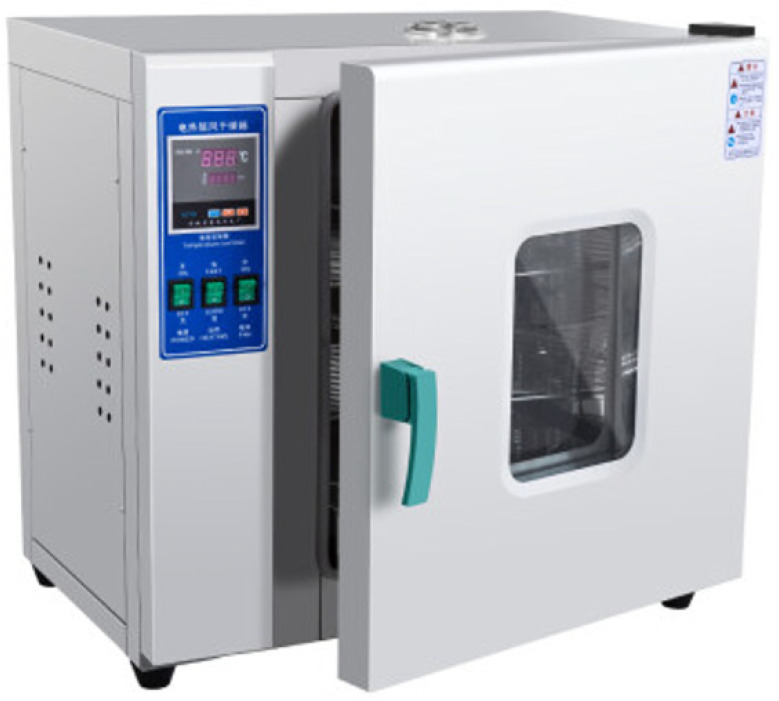
As shown here, 101-2B electric blast constant temperature drying oven.

**Figure 5 polymers-15-02209-f005:**
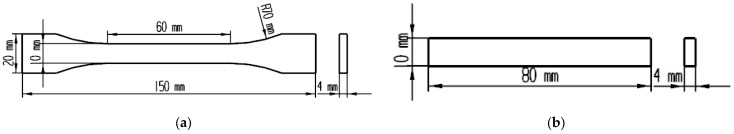
(**a**) Size of tensile samples; (**b**) size of bending samples.

**Figure 6 polymers-15-02209-f006:**
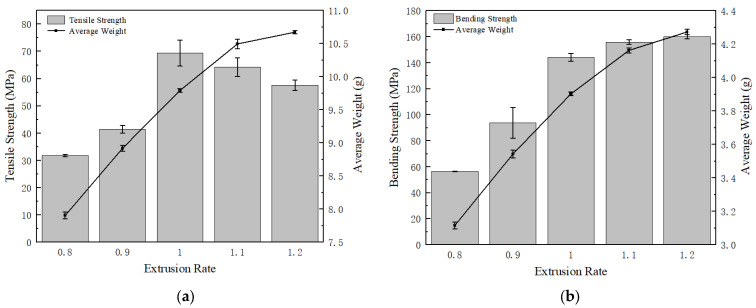
(**a**) Tensile test results with different extrusion rates; (**b**) bending test results with different extrusion rates.

**Figure 7 polymers-15-02209-f007:**
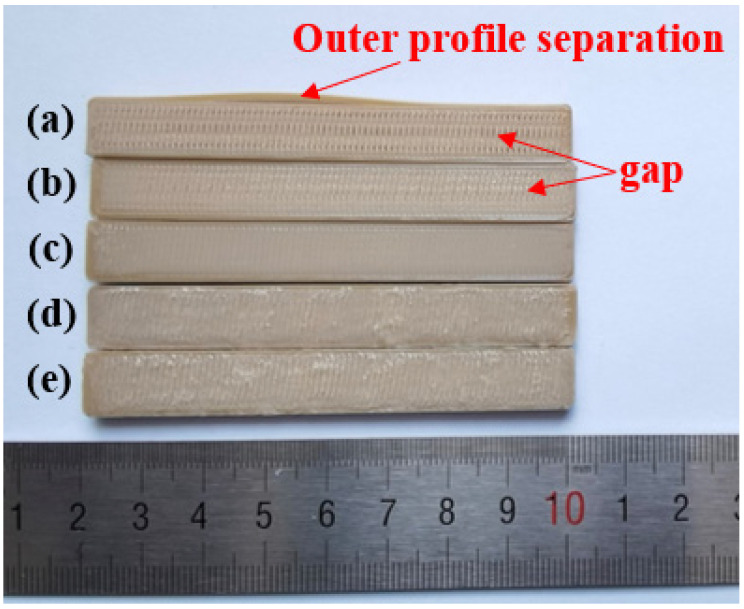
Bending specimens from different extrusion rates: (**a**) 0.8 times; (**b**) 0.9 times; (**c**) 1.0 times; (**d**) 1.1 times; (**e**) 1.2 times.

**Figure 8 polymers-15-02209-f008:**
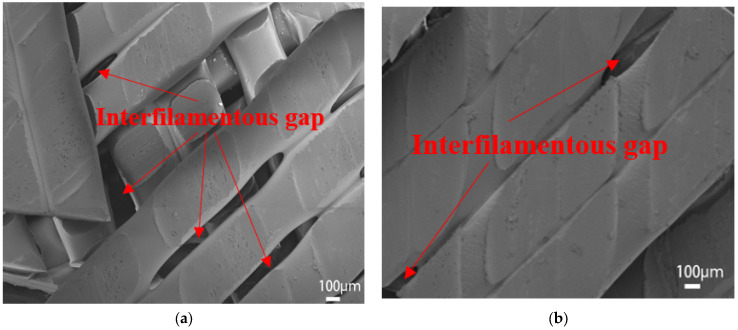
(**a**) Magnified SEM (scanning electron microscopy) images for the 0.8 times extrusion rate; (**b**) magnified SEM images for the 0.9 times extrusion rate.

**Figure 9 polymers-15-02209-f009:**
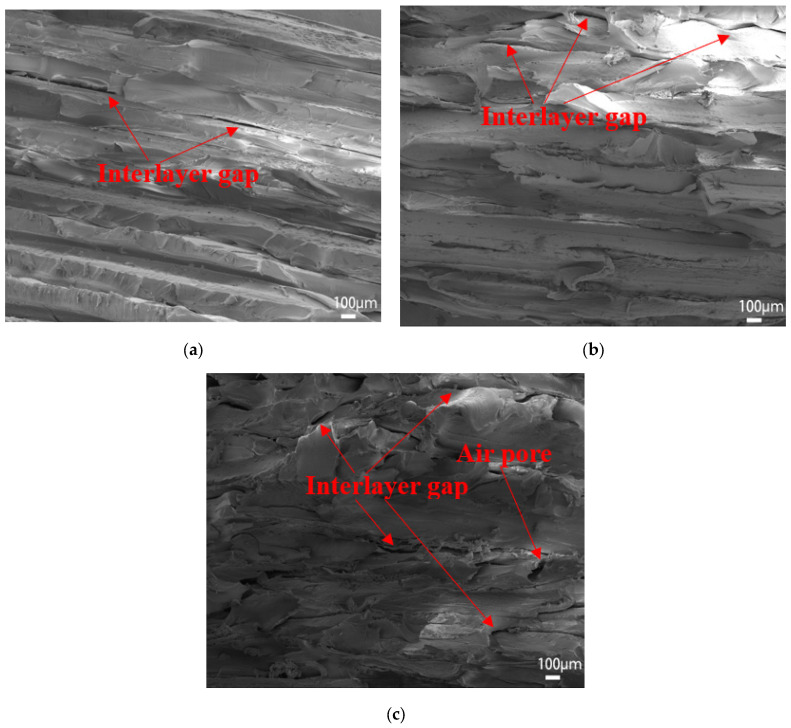
(**a**) Magnified SEM images for the 1.0 times extrusion rate; (**b**) magnified SEM images for the 1.1 times extrusion rate; (**c**) magnified SEM images for the 1.2 times extrusion rate.

**Figure 10 polymers-15-02209-f010:**
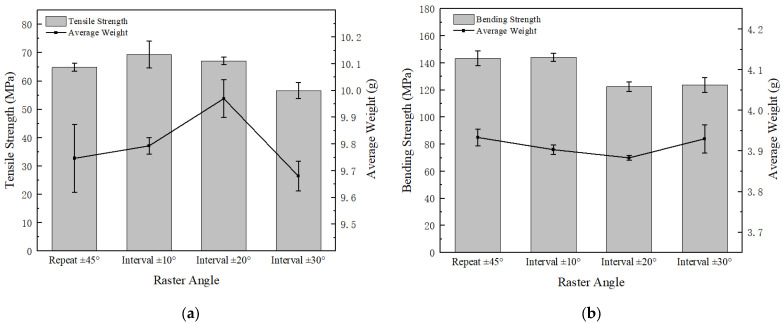
(**a**) Tensile test results of differently filled samples; (**b**) bending test results of differently filled samples.

**Figure 11 polymers-15-02209-f011:**
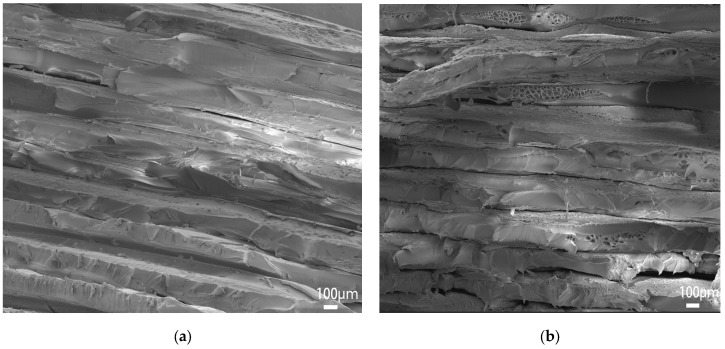
(**a**) Microscopic SEM images for cross-sections from cross-infilled samples at varied angle intervals of ±10°; (**b**) microscopic SEM images for cross-sections from cross-infilled samples at varied angle intervals of ±20°; (**c**) microscopic SEM images for cross-sections from cross-infilled samples at varied angle intervals of ±30°.

**Figure 12 polymers-15-02209-f012:**
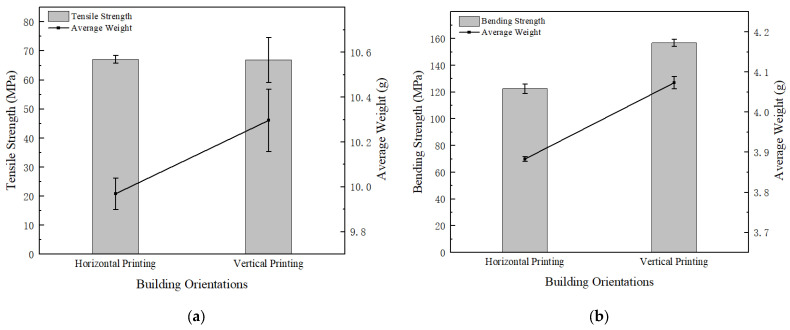
(**a**) Tensile test results of different printing orientations; (**b**) bending test results of different printing orientations.

**Figure 13 polymers-15-02209-f013:**
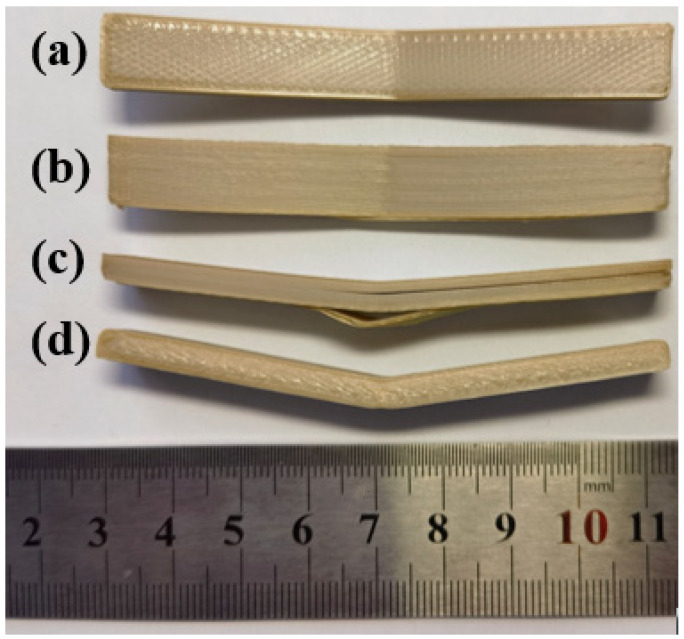
Three-point bending test failure samples: (**a**) horizontal print sample; (**b**) vertical print sample; (**c**) interlayer debonding of horizontal printing sample; (**d**) failure details of vertical printing sample.

**Figure 14 polymers-15-02209-f014:**
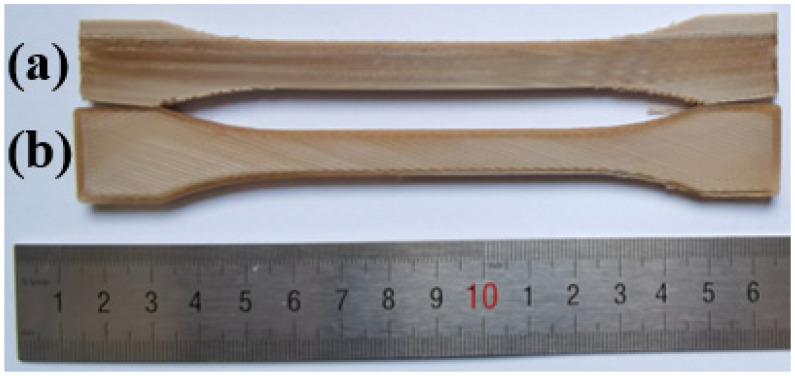
PEEK parts with different printing orientations: (**a**) vertical printing tensile sample; (**b**) horizontal printing tensile sample.

**Figure 15 polymers-15-02209-f015:**
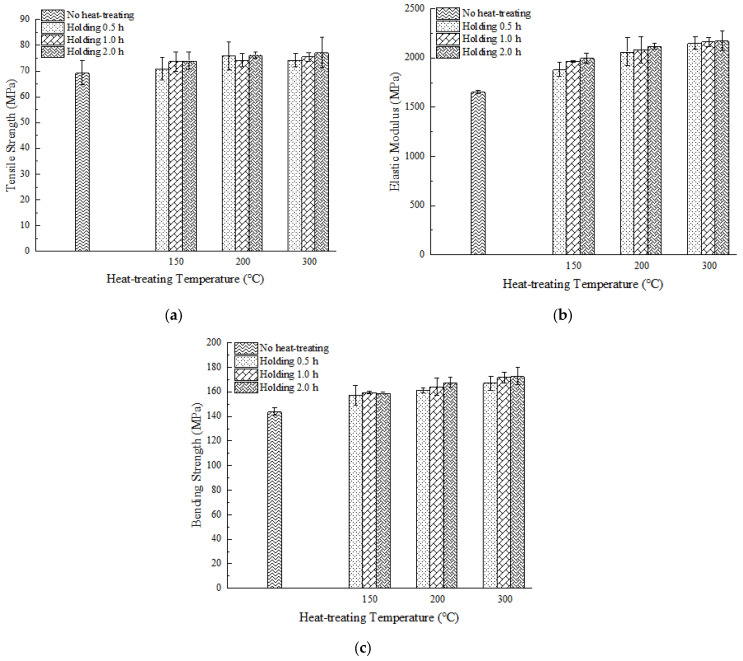
(**a**) Tensile strength test results of heat treatment samples; (**b**) tensile modulus test results of heat treatment samples; (**c**) bnding strength test results of heat treatment samples.

**Figure 16 polymers-15-02209-f016:**
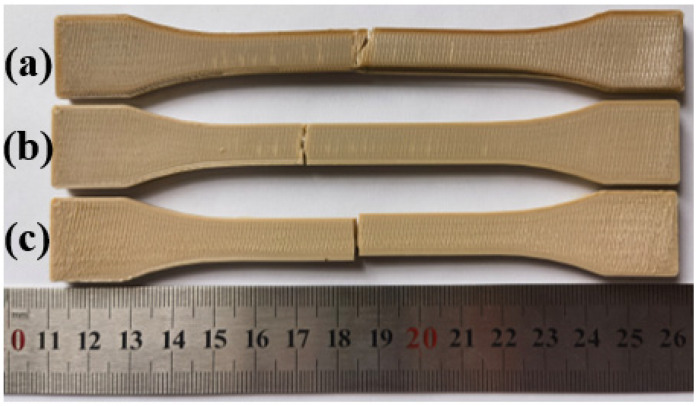
Tensile fracture samples with different heat treatment temperatures: (**a**) 150 °C; (**b**) 200 °C; (**c**) 300 °C.

**Figure 17 polymers-15-02209-f017:**
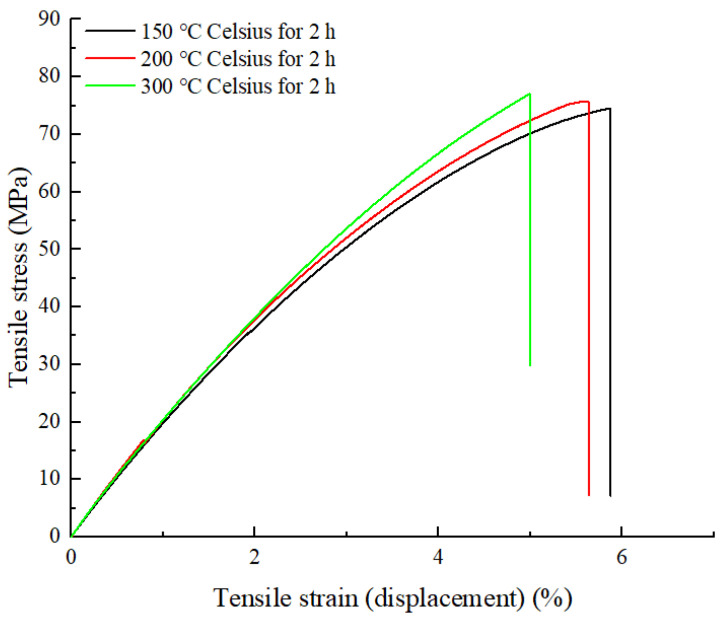
Stress–strain curves of tensile samples with different heat treatment temperatures.

**Table 1 polymers-15-02209-t001:** Printing parameter settings.

Print Parameters	Fill Angle	Extrusion Rate	Print Orientation
Printing temperature: 430 °C Printing speed: 2400 mm/min Nozzle diameter: 0.4 mm Layer thickness: 0.2 mm No auxiliary heat source	Interval ±10°	0.8 times	Horizontal printing
		0.9 times	
		1.0 times	
		1.1 times	
		1.2 times	
	Interval ±20°	1.0 times	Horizontal printing
			Vertical printing
	Interval ±30°	1.0 times	Horizontal printing

**Table 2 polymers-15-02209-t002:** Heat treatment parameter settings.

Heat Treatment Parameters	Temperature/°C	Time/h
After the oven is heated to the specified temperature, the PEEK parts are placed in the oven and left to cool in the air for a predetermined time.	150	0.5 1 2
	200	0.5 1 2
	300	0.5 1 2

**Table 3 polymers-15-02209-t003:** Extreme values of mechanical properties with different printing parameters.

	Test Result	Tensile Strength		Improvement	Bending Strength		Improvement
Printing Parameters		Worst	Best		Worst	Best	
Extrusion rate		0.8 times /31.78 MPa	1.0 times /69.35 MPa	118.22%	0.8 times /56.48 MPa	1.2 times /160.88 MPa	183.43%
Filling angle		±30°/56.65 MPa	±10°/69.35 MPa	22.42%	±20° /122.53 MPa	±10° /144.16 MPa	17.65%
Print orientation		Vertical /66.97 MPa	Horizontal /67.14 MPa	0.25%	Horizontal /122.53 MPa	Vertical /156.84 MPa	28.00%

**Table 4 polymers-15-02209-t004:** Extreme values of mechanical properties with different heat treatment parameters.

	Test Result	Tensile Strength		Improvement	Bending Strength		Improvement
Printing Parameters		Worst	Best		Worst	Best	
Temperature (0.5 h)		150 °C /70.84 MPa	300 °C /74.24 MPa	4.80%	150 °C /157.37 MPa	300 °C /167.13 MPa	6.20%
Time (300 °C)		0.5 h/74.24 MPa	2 h/77.26 MPa	4.07%	0.5 h /167.13 MPa	2 h/172.98 MPa	3.50%

**Table 5 polymers-15-02209-t005:** Crystallinity test results.

Parameters		
Temperature/°C	Time/h	Crystallinity
150	0.5	18.03%
	1.0	22.98%
	2.0	23.77%
200	0.5	23.49%
	1.0	23.75%
	2.0	24.64%
300	0.5	22.14%
	1.0	26.45%
	2.0	28.70%
Without heat treatment		16.10%

## Data Availability

Not applicable.

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
