# Peer review of "Effect of 3D Printing Process Parameters and Heat Treatment Conditions on the Mechanical Properties and Microstructure of PEEK Parts"

_polymers, 2023, doi:10.3390/polym15092209_

Round 1
Reviewer 1 Report
Dear Authors,
Thank you for this very interesting work related to PEEK material, its material properties, and how they can be improved by adjusting the process variables as well as by selecting the appropriate heat treatment (temperature, time) settings. Overall, the work is well-defined, however, there are some comments that have to be addressed.
Abstract
The first two sentences should be considered as part of the introduction section as it seems that contain duplicate material.
This section could begin with the challenges that are related to the methodology of the present work or the challenges of the end part which have been made with AM (From However and below).
Additionally, please try to include in which applications these materials are used and what kind of properties are required.
Introduction
It is true that FDM process and the relative materials are used more and more in functional applications. Apart from PEEK, reinforced polymers are used in structural applications. Apart from the strength, the decision-making includes the printability as well as the post-processing steps that are related to each material, including the required specifications from the machine such as heated bed, closed build chamber, etc. To this end, the authors should include relative references that point out the characteristics of materials that are used for functional use and their limitations/challenges.
Please find below two indicative works that contain such information and can be included in the work. The first one introduces the different steps that need to be considered during decision making, relative to material-process selection, while the second one represents the challenges of the materials introduced in FDM process.
· H. Bikas, N. Porevopoulos, P. Stavropoulos, "A decision support method for knowledge-based Additive Manufacturing process selection", CIRP Manufacturing Systems Conference 2021, Volume 104, pg. 1650-1655 , 22-24 September 2021, Athens, Greece (2021)
· P. Stavropoulos, P. Foteinopoulos, "Modelling of Additive Manufacturing Processes: A Review and Classification", Manufacturing Review, Vol. 5, No. 2, pg. 8-34, (2018)
Additionally, neither during the introduction section nor on any other section the effect of layer height is discussed. Based on the following work, the layer height affects both the quality of the end part, the build time as well as the structural properties. Please include this reference in your work, discussing on the effect of layer height on the process.
· Wu, Wenzheng & Geng, Peng & Li, Guiwei & Zhao, Di & Zhang, Haibo & Zhao, ji. (2015). Influence of Layer Thickness and Raster Angle on the Mechanical Properties of 3D-Printed PEEK and a Comparative Mechanical Study between PEEK and ABS. Materials. 8. 5834-5846. 10.3390/ma8095271.
Moreover, please introduce the materials PEEK, ABS, etc when they are mentioned for first time in this section.
Finally, please discuss if the outputs from the tensile and bending testing for the standard dumbbell artefacts are generalizable, considering the anisotropy of material and the load paths from a specific case, and include a specific case study to test the material properties.
Experimental preparation
How the extrusion rate is defined? Moreover, why it has been used as ratio of velocities and not two separate speeds such as wire speed and printer head speed?
As regards the design of the dumbbell, why the authors did not test different scales of the material to see if the overall dimensions of a part made from AM processes could affect the results, considering different heat accumulation and cooling time between the layers.
Experimental results and discussion
As a section, it is well organized and divided in subsections so as to represent in an accurate way the outputs. Although the results can be understood, the effect of the different process parameters on the key performance indicators are not very well presented. It seems that a statistical analysis is required based on the given datasets in order to clarify which process variable may affect the most the process compared to the others.
Additionally, the results about warpage and mechanical properties cannot be related with the material but maybe to the specific design and the material. Somehow these two variables have to be distinguished.
Please find the following work that is related to another AM process, where the process variables affect the defects of the part on the different layers. However, these defects may change when different path planning strategies are implemented with the same input material and speed, due to the different cooling time, which is time, what is needed so as the head to pass from the same point consecutive times.
· P. Rey, C. Prieto, C. González, K. Tzimanis, T. Souflas, P. Stavropoulos, J. S. Rathore, V. Bergeaud, C. Vienne, P. Bredif, "Data Analysis to asses part quality in DED-LB/M based on in-situ process monitoring", 12th CIRP Conference on Photonic Technologies [LANE 2022], Vol. 111, pg. 345-350 , 4-8 September, Fürth, Germany (2022)
Conclusions and future work
Well-presented but not completely supported by the results. Statistical analysis could improve the demonstration of results while from a technical perspective the content is fine.
Thank you
Reviewer 2 Report
The study presents the correlation of the printing parameters and the post heat treatment to the mechanical properties of the FDM PEEK. A few comments as below:
1. The machine used to manufacture could significantly affect the properties of the PEEK. Which machine(model) did the author use?
2. please clearly define the 1.2 times (or others) extrusion rate. Does that mean the extruded materials are expected to cover 1.2 times the width of the path? How much does PEEK shrink after the heating-cooling cycle?--that affects the extrusion rate, too.
3. warpage effect: One can usually use a brim or raft to overcome this effect. Would the authors expect or try to see if adding a brim or raft could solve the warpage?
4. Would the authors expect the heat treatment to heal the defects, besides increasing the crystallinity?
Reviewer 3 Report
1. Full names must be provided before abbreviations can be used, such as ABS, PLA, PC, etc.
2. In the introduction, the authors summarized the investigations on the effects of printing parameters. Please combine 3-6 paragraphs into one paragraph.
3. Any limitations of current studies on the effects of heat treatment? The authors summarized the related articles but it was still difficult to understand why it's necessary to separately study the effects of heat treatment parameters.
4. Important information was needed in the experimental preparation. For example, which FDM 3D printer was used for printing? How to run heat treatment? Which device? How to run tensile and bending tests? Devices? Details in SEM and DSC?
5. Some figures can be combined together, e.g., Figures 2 and 3, Figures 4 and 5, Figures 6 and 7, etc. Otherwise, there were too many figures in the manuscript.
6. There were no discussions in Section 3. The authors just described the trends and phenomena in each figure without providing explanations. For example, why 1.0 times extrusion rate, 252 vary-angle cross-fillings with ±10° filling intervals and vertical printing were the best combination? Any specific reasons? Also, did other researchers observe the similar phenomenon? Any citations?
7. Figures 18-20, please use different patterns to distinguish different heat-treatment conditions in each figure.
8. How to measure crystallinity? Was this information provided in detail in Section 2?
9. More discussions were needed. For example, in Figures 21 and 22, the strength of the sample parts after heat treatment all increased regardless of the processing temperatures, but the toughness deteriorated at the same time, why? Why no necking stage was observed and only brittle fracture occurred? Any explanation?
10. More citations were needed.
Reviewer 4 Report
The manuscript entitled " Effect of 3D printing process parameters and heat treatment conditions on the mechanical properties and microstructure of PEEK parts" investigated the effect of the printing parameters e.g., extrusion rate, filling angle, and printing orientation on the mechanical properties of 3D printed PEEK parts. The optimized parameters are provided as well.
My comments are listed below:
1. Can the author elaborate on how the crystalline improves the layer-layer bonding strength? Is it possible to conclude that after heat treatment, the increased crystallinity is mostly on the bonding area?
2. Are Fig 9 (a) and (b) from the same location? If so, can the authors mark the area of Fig 9(b) on Fig.9 (a)? Moreover, seems Fig.9 (b) won't provide any additional information.
3. As the result shown in Fig.11, can the authors elaborate more on why the interval +/- 10 degrees shows the largest variation on the tensile strength while others are relatively smaller?
4. Fig.21 shows the breaking point is not around the center region of the dogbone structure. It's concerned it will induce some bias in the uniaxial tension test result. Can the authors repeat the tension experiment with improved sample quality?
Round 2
Reviewer 1 Report
All comments have been addressed. It can be published in its current form.
Reviewer 3 Report
The authors updated the manuscript greatly based on my comments. Thank you for the efforts. I believe the paper was ready to be published.